

# ScintPi measurements of low-latitude ionospheric irregularity drifts using the spaced-receiver technique and SBAS signals

Josemaria Gomez Socola[1], Fabiano S. Rodrigues[1], Isaac G. Wright[1], Igo Paulino[2] and Ricardo Buriti[2]

[1]W.B. Hanson Center for Space Sciences, The University of Texas at Dallas, Dallas, 75080, United States
[2]Department of Physics, Federal University of Campina Grande, Campina Grande, 58429-900, Brazil

*Correspondence to*: Josemaria Gomez Socola (Josemaria.GomezSocola@utdallas.edu)

**Abstract.** Previous efforts have used pairs of closely-spaced specialized receivers to measure Global Navigation Satellite System (GNSS) signals and to estimate ionospheric irregularity drifts. The relatively high cost associated with commercial GNSS-based ionospheric receivers limited somewhat their deployment and the estimation of ionospheric drifts. The development of an alternative low-cost GNSS-based scintillation monitor (ScintPi) motivated us to investigate the possibility of using it to overcome this limitation. ScintPi monitors can observe signals from geostationary satellites, which can greatly simplify the estimation of the drifts. We present results of an experiment to evaluate the use of ScintPi 3.0 to estimate ionospheric irregularity drifts. The experiment consisted of two ScintPi 3.0 deployed in Campina Grande, Brazil (7.213°S, 35.907°W, dip latitude ~14°S). The monitors were spaced by 140 m in the magnetic east-west direction and targeted the estimation of the zonal drifts associated with scintillation-causing equatorial spread F (ESF) irregularities. Routine observations throughout an entire ESF season (September 2022 – April 2023) were made as part of the experiment. We focused on results of irregularity drifts derived from geostationary satellite signals. The results show that the local time variation in the estimated irregularity zonal drifts is in good agreement with previous measurements and with the expected behavior of the background zonal plasma drifts. Our results also reveal a seasonal trend in the irregularity zonal drifts. The trend follows the seasonal behavior of the zonal component of the thermospheric neutral winds as predicted by the Horizontal Wind Model (HMW14). This is explained by the fact that low latitude ionospheric F-region plasma drifts are controlled, in great part, by Pedersen conductivity weighted flux tube integrated zonal neutral winds. The results confirm that ScintPi has the potential to contribute to new, cost-effective measurements of ionospheric irregularity drifts in addition to scintillation and total electron content. Furthermore, the results indicate that these new ScintPi measurements can provide insight on ionosphere-thermosphere coupling



## 1 Introduction

A series of low-cost, easy to install and easy to maintain ionospheric scintillation and total electron content (TEC) monitors
were developed at UT Dallas (e.g., Rodrigues and Moraes, 2019; Gomez Socola and Rodrigues, 2022). These monitors are
referred to as ScintPi and were developed using commercial off-the-shelf (COTS) Global Navigation Satellite System (GNSS)
receivers and single-board computers (Raspberry Pi). While ScintPi monitors were not developed to fully replace commercial
monitors, they can be used in various scientific and educational applications. For instance, ScintPi monitors have been
successfully used in studies of low and low-to-mid latitude ionospheric irregularities and TEC depletions (e.g., Sousasantos et
al., 2023,2024; Gomez Socola et al., 2023), in studies of solar radio bursts and their impact on GNSS signals (Wright et al.,
2023a) and in student projects related to space weather (Wright et al., 2023b).

Given that ScintPi has a reduced sampling rate (20 Hz) and C/No resolution (1 dB-Hz) compared to typical commercial
scintillation monitors, the ability of ScintPi to produce satisfactory drift estimates needed to be investigated. On the other hand,
ScintPi is capable of monitoring signals transmitted by geostationary satellites, which would greatly simplify the analyses
associated with and interpretation of irregularity drift measurements.

Therefore, here we present results of an experimental effort to evaluate the use of ScintPi monitors to estimate ionospheric
irregularity drifts. The approach follows that of the so-called closely-spaced scintillation technique which measures time
differences between the occurrence of scintillation patterns observed by closely-spaced receivers to infer the horizontal
velocity component of scintillation-causing irregularities along the receivers baseline.

Ionospheric plasma drifts represent an important feature of the ionospheric plasma behavior and are responsible for controlling
plasma transport and structuring. For instance, vertical (upward) F-region plasma drifts drive the transport of plasma from the
magnetic equator to low latitudes through the ionospheric plasma fountain effect (Hanson and Moffett, 1966; Moffett, 1979).
This mechanism gives rise to the Equatorial Ionization Anomaly – EIA (Appleton, 1946; Bailey 1948). Additionally, the so-
called pre-reversal enhancements (PRE) in vertical drifts, occurring near sunset, is known to control the development of
equatorial spread F – ESF (Heelis et al., 1974; Farley et al., 1986; Fejer et al., 1999; Fesen et al., 2000).

The magnetic zonal component of the F-region plasma velocity at low latitudes, on the other hand, can reflect the level of
coupling between the upper thermosphere and ionosphere (Immel et al., 2006; Wang et al., 2021). During nighttime, the E-
region dynamo is greatly reduced, and F-region plasma drifts should reflect the effects of the F-region dynamo alone (e.g.,
Coley et al., 1994).

Ionospheric plasma drifts are also relevant in certain applications. According to Carrano and Groves, (2010), the probability of experiencing loss of lock and likelihood of extended signal reacquisition times depend on factors such as the velocity of the satellite motion with respect to the geomagnetic field and plasma drifts.

Obtaining ground-based measurements of background ionospheric plasma drifts presents significant challenges. For instance, 65  the incoherent scatter radar (ISR) of the Jicamarca Radio Observatory (JRO) is the only instrument capable of making observations of vertical and zonal plasma drifts. While the measurements can be conducted across various heights within the F-region, the observations require the operation of high-power transmitters which limit the number of measurements per year.

The motion of ionospheric irregularities can be measured more easily than the motion of the background ionospheric plasma 70  using, for instance, L-band spaced scintillation receivers.(Kil et al., 2000,2002; Ledvina et al., 2004; Kintner et al., 2004; Kintner and Ledvina, 2005; Cerruti et al., 2006; Otsuka et al., 2006; Sobral et al., 2009; de Paula et al., 2010; Muella et al., 2008,2009,2012,2014,2017; Cesaroni et al., 2021). Irregularity drifts are often used as an indicator of background plasma drifts at low latitudes. The application of the scintillation-based spaced receiver technique to measure irregularity drifts is revisited in this study with focus on using more cost-effective instrumentation for the measurements.


Gomez Socola and Rodrigues (2022) already addressed the ability of the sampling rate (20 Hz) and amplitude resolution (1 dB-Hz) of ScintPi 3.0 measurements to produce amplitude scintillation indices that are comparable to those produced by commercial GNSS-based monitors. Here, we present results of an experimental setup and campaign that targeted evaluating the ability of applying the scintillation-based spaced receiver technique to ScintPi 3.0 measurements and estimate irregularity 80  drifts.

The presentation of our work is organized as follows: In Sect. 2 we provide information about the ScintPi 3.0, measurements, spaced-receiver experimental setup and geophysical conditions under which measurements were made. Sect. 3 describes the methodology used to derive irregularity drifts from the ScintPi measurements. In Sect. 4 we present and discuss the results of 85  the measurements and analyses. Sect. 5 summarizes our main findings.

## 2 Experimental setup

Scintillation observations were carried out using ScintPi 3.0 monitors. ScintPi 3.0 can be described as a multi GNSS constellation, dual-frequency ionospheric scintillation and total electron content (TEC) monitor. ScintPi was developed at UT Dallas using a COTS GNSS receiver and a single-board computer (Raspberry Pi).




The development of ScintPi was motivated by the relatively high cost of commercial GNSS-based scintillation and TEC monitors. While it is not intended to fully replace commercial monitors, ScintPi allows different research and educational initiatives. Details about ScintPi 3.0, along with illustrative examples of measurements and comparison with collocated observations made by commercial receivers are described in Gomez Socola and Rodrigues (2022). Examples of different types of studies using ScintPi measurements can be found in Gomez Socola et al. (2023), Sousasantos et al. (2023; 2024) and Wright et al. (2023a). Results of an educational effort using ScintPi can be found in Wright et al. (2023b). Of relevance to this study is that ScintPi can measure the strength of signals transmitted by GNSS satellites, achieving sampling rates of 20 Hz and C/No resolution of 1 dB-Hz. In this study, our focus centers on the analysis of signals transmitted by geostationary satellites. The use of geostationary satellites simplifies the analyses and the interpretation of the measurements.

## 2.1 Deployment and spacing

Because low-latitude equatorial plasma bubbles (EPBs) and ionospheric irregularities associated with them are known to move in the magnetic zonal direction (Dyson and Benson, 1978; Mendillo and Baumgardner, 1982; Martinis et al., 2003, 2020), the two monitors were placed along the east-west magnetic direction spaced by 140 m. The site is in a region of -21.4° magnetic declination based on magnetic values from the International Geomagnetic Reference Field - IGRF-13 (Alken et al., 2021) (See Fig. 1b). The spacing was chosen based on expected values of low latitude irregularity drifts, sampling rate of the receivers, and availability of locations. Figure 1a shows the location of the observation site (UFCG). Figure 1b and 1c describe the placement of the receivers.

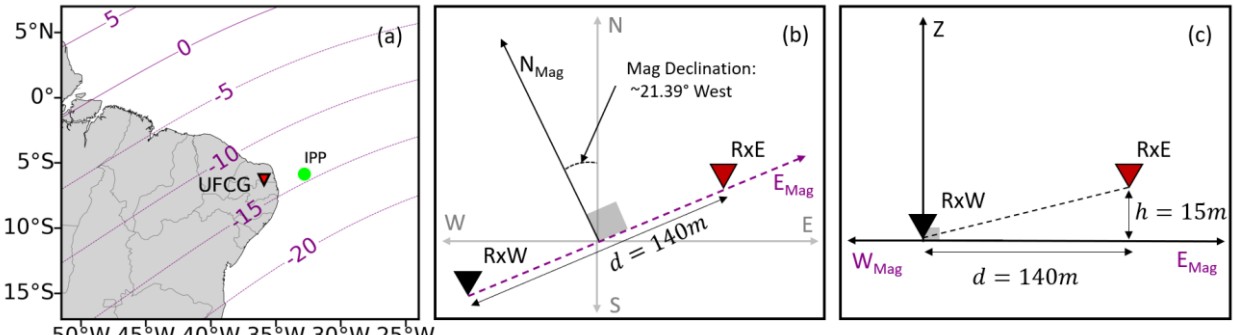

**Figure 1: (a) Map showing the location of UFCG where spaced ScintPi 3.0 monitors have been deployed for this study. UFCG is located at ~14° dip latitude where L-Band scintillations occur frequently. Panels (b) and (c) are diagrams showing the placement of the monitors along the magnetic zonal direction for estimation of ionospheric irregularity drifts using the scintillation-based spaced-received technique**

## 2.2 Measurements

While ScintPi 3.0 can make measurements of signals transmitted by different GNSS satellites (GPS, GLONASS, Beidou, and Galileo), in this study we focused on measurements of signals transmitted by geostationary satellites which can also be measured by ScintPi 3.0. The use of geostationary signals has various advantages. For instance, signals from geostationary satellites provide observations over a fixed location allowing for unambiguous interpretation of the variability in the observed



drifts. Additionally, the decorrelation times of the observed signal fluctuations are not affected by satellite velocities. Perhaps more importantly, the satellite velocity does not have to be considered in the analyses and calculations of ionospheric
irregularity drifts.

The signals recorded by GNSS receivers are coded using pseudo-random noise (PRN) sequences, which are codes that each satellite transmits to differentiate itself from other satellites. At the location of UFCG, ScintPi 3.0 monitors can decode signals from the Astra-5B (PRN123) and Astra Ses-5 (PRN136) geostationary satellites, stationed at 31.5°E and 5°E geographic
longitudes, respectively. Unfortunately, data from PRN123 had to be excluded due to its low elevation angle (~14°) which led to non-geophysical amplitude fading caused by multipath. PRN136, on the other hand, was positioned at an elevation angle of 42° and azimuth of 84° which provided adequate measurements for this study. The ionospheric pierce point (IPP) for the PRN136 signal is located at 6.77°S, 31.86°W (dip latitude ~15°S), as shown in Fig. 1a. The IPP calculation followed the method outlined by Prol et al. (2017), considering the altitude of the ionospheric F-region peak density at 450 km.

For this study, we analyzed 8 months (September 1, 2022 – April 30, 2023) of nearly continuous measurements. The measurements covered an entire ESF season in Brazil when scintillations occur frequently. The measurements were made during the ascending phase of solar cycle 25, and the mean solar flux for the period was 148.23 SFU.

**3 Scintillation pattern velocities and ionospheric irregularity drifts**

As previously mentioned, ScintPi 3.0 measures the strength of L-band signals transmitted by geostationary satellites. These signals are subject to diffraction by irregularities in the ionospheric plasma density, giving rise to intensity fading (scintillation) patterns observed by monitors on the ground.

At low magnetic latitudes, ionospheric irregularities are known to be elongated along the magnetic field lines and to drift in
the zonal magnetic direction (Dyson and Benson, 1978; Mendillo and Baumgardner, 1982; Martinis et al., 2003, 2020). Therefore, an approach to estimate zonal irregularity drift is to measure the scintillation pattern velocity using receivers that are spaced in the magnetic zonal direction.

An experimental setup commonly used to measure the scintillation pattern velocity relies on the estimation of the time delay
between scintillation patterns observed by spaced scintillation receivers. This time delay is quantified by the time lag at which the cross-correlation function of the measured signals reaches its maximum ($\tau_0$). The measured (or apparent) scintillation velocity along the receiver baseline ($v'_{scint}$) is then quantified from $v'_{scint} = d/\tau_0$, where $d$ represents the spacing between the receivers (Kil et al., 2000).



In this study, we take into consideration the approach introduced by Briggs et al. (1950) which compensates for the temporal evolution of the intensity pattern. This approach is known as the "full correlation method" and has been widely used in estimation of irregularity drifts (e.g., Kil et al., 2000; Kintner et al., 2004; Kintner and Ledvina, 2005; Otsuka et al., 2006; Sobral et al., 2009; de Paula et al., 2010; Muella et al., 2008,2009,2012,2014,2017). We summarize the approach in the following paragraphs.


Briggs et al. (1950) pointed out that the apparent velocity ($v'_{scint}$) has contribution from the random-like evolution of scintillation-causing irregularities as they move between the line-of-sight of the two observing receivers. The apparent velocity can be related to the true velocity ($v_{scint}$) through the following expression (Briggs et al.,1950):

$$v_{scint} = \frac{v'_{scint}}{1+t_0^2/\tau_0^2}, \tag{1}$$

Where $t_0$ is the so-called characteristic time and is the time at which the autocorrelation of the scintillation pattern matches the maximum cross-correlation value. This time is associated with a random component in the motion of the plasma density irregularities and, therefore, the ratio $t_0^2/\tau_0^2$ determines the extent to which the apparent and the true velocity are equivalent. For instance, when random motion (decorrelation) is negligible, the cross-correlation value for the two scintillation patterns approaches 1. As a result, $t_0$ will approach 0 and the ratio $t_0^2/\tau_0^2$ will be negligible. The measured velocity would then match

the true velocity ($v_{scint} \sim v'_{scint}$). In contrast, situations where $t_0$ and $t_0^2/\tau_0^2$ depart from 0 indicate that the scintillation pattern velocity has non-negligible contributions from random motion (Kintner et al., 2004). Finally, to quantify the contribution from random motions to the observed velocities, Briggs et al. (1950) introduced the so-called characteristic velocity ($v_c$):

$$v_c = v_{scint}\left(\frac{t_0}{\tau_0}\right) = \frac{v'_{scint}}{1+t_0^2/\tau_0^2}\left(\frac{t_0}{\tau_0}\right), \tag{2}$$

Combining Eq. (1) and Eq. (2) one can write:

$$v'_{scint} = v_{scint} + \frac{v_c^2}{v_{scint}}, \tag{3}$$

Equation 3 shows, more explicitly, that the measured pattern velocity ($v'_{scint}$) has contributions from the true irregularity drifts and from the random characteristic irregularity velocity.

Figure 2 illustrates the estimation of $\tau_0$ and $t_0$ used to compute true velocities (Eq. 1) and characteristic velocities (Eq. 2).

Figure 2a shows a 60-second interval of signal strengths measured by the receivers at UFCG. Note that, despite the low C/No resolution (1 dB-Hz), similar fluctuations are observed by both receivers. Also note that fluctuations observed by the west receiver (RxW) precede the fluctuations observed by the east receiver (RxE) indicating eastward motion. Figure 2b shows the cross-correlation function between these two signals. From the maximum cross-correlation coefficient we obtain the time delay ($\tau_0$) needed to compute the apparent pattern velocity ($v'_{scint}$). Figure 2c shows the autocorrelation of the scintillation pattern



measured by the west receiver. The maximum cross-correlation coefficient is used to estimate the characteristic time ($t_0$) (see gray line in panels b and c). Again, we point out that, for high cross-correlation values (say correlation coefficient $\sim 1$), $t_0$ and $t_0^2/\tau_0^2$ would have negligible values, and the apparent velocity ($v'_{scint}$) , Eq. (1) would converge to the true irregularity velocity ($v_{scint}$).

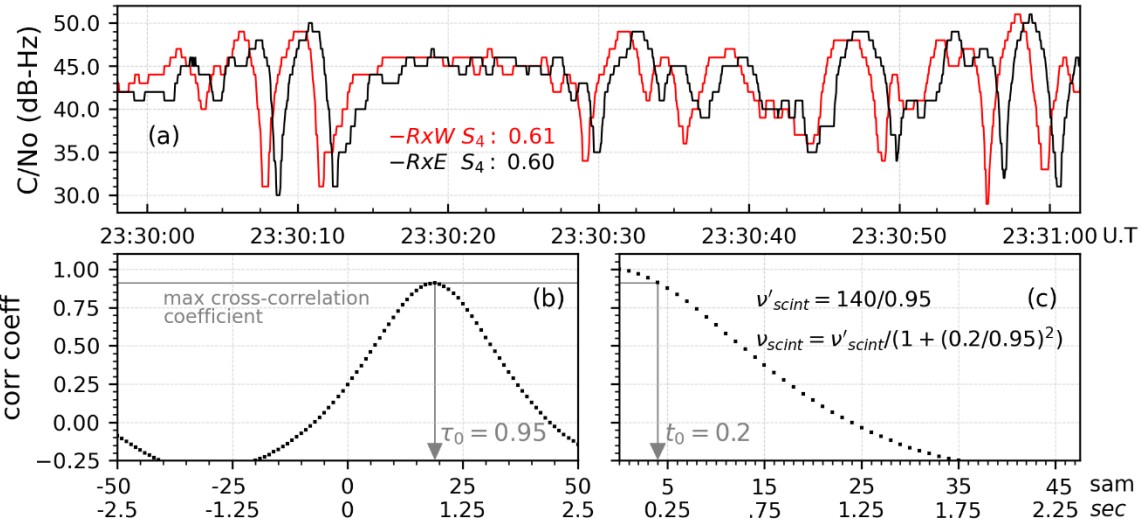

Figure 2 – **Illustration of calculation used to derive the characteristic time ($t_0$) and the time lag of maximum correlation ($\tau_0$). Panel**
**(a) shows an example of 60-second signal strength measurements made with closely spaced ScintPi 3.0 monitors on October 4, 2022. Panel (b) shows the cross-correlation function between power series from the two receivers, and panel (c) shows the autocorrelation for the west receiver. The maximum cross-correlation coefficient is used to estimate the characteristic time associated with the signal decorrelation. The horizontal axis in panels (b) and (c) show both sample number (samp) and time in seconds.**

### 3.1 Further considerations

While most studies (e.g., Kil et al., 2000; Otsuka et al., 2006; de Paula et al., 2010) have used the Brigg's formulation (Eq. 1), Ledvina et al. (2004) presented further analysis of the spaced-receiver technique and pointed out additional considerations. First, they pointed out the fact that the scintillation pattern velocity is a projected representation of the velocity of the irregularities, and this velocity will depend, in general, on geometric factors involving the placement of the receivers and the Earth's magnetic field. For instance, differences in the altitudes of the two receivers should be taken into consideration in the

estimation of the drifts. They showed that Eq. (1) can be updated to take into consideration differences in the heights of the receivers (Ledvina et al., 2004):

$$v_{scintx} = \left[1 - \frac{\Delta h}{d}\tan(\theta_{vz})\right]\left[\frac{v'_{scintx}}{1+t_0^2/\tau_0^2}\right], \tag{4}$$



Where $v_{scintx}$ represents the scintillation pattern velocity in the east-west direction parallel to the Earth's surface, $d$ is the zonal spacing of the receivers, $\Delta h$ is the height difference of the receivers, and $\theta_{vz}$ is a projection angle in the vertical-zonal plane. The projection angle is given by (Ledvina et al., 2004):

$$\theta_{vz} = \tan^{-1}\left[\frac{\tan(\phi)\sin(\theta) - (B_x/B_y)\tan(\phi)\cos(\theta)}{1 - (B_z/B_y)\tan(\phi)\cos(\theta)}\right], \tag{5}$$

Where $\phi$ is the satellite zenith angle and $\theta$ is the satellite azimuth angle from magnetic north. $B_x$, $B_y$, and $B_z$ are components of geomagnetic field vector at the IPP location in the magnetic east, north and vertical directions, respectively. The IPP was computed using a mean ionospheric altitude of 450 km.

Using long-term averages of the GNSS coordinates provided by the receivers we found a height difference of 15 meters (Fig. 1c). Taking into consideration values for the experiment at UFCG, the expression for the irregularity drifts using the geostationary measurements is updated to:

$$v_{scintx} = [0.899]\left[\frac{v'_{scintx}}{1 + t_0^2/\tau_0^2}\right], \tag{6}$$

Ledvina et al. (2004) also pointed out that deriving zonal irregularities drifts from the scintillation pattern velocity can be affected by vertical ionospheric drifts ($v_{iz}$) at the IPP location. The vertical drifts would affect the scintillation pattern velocity as follows:

$$v_{scintx} = v_{zonal} + \left[\frac{-\tan(\phi)\sin(\theta)}{1 - (B_z/B_y)\tan(\phi)\cos(\theta)}\right]v_{iz}, \tag{7}$$

Where $(B_z/B_y)$ is equivalent to the tangent of inclination angle of the geomagnetic field ($(\tan(\vartheta_{dip}))$) at the IPP location, and $v_{iz}$ is the vertical irregularity drift.

For our experimental setup, $\phi = 48°$, $\theta = (84°+21.39°)$, and $\vartheta_{dip} = -29.62°$ resulting in $v_{scintx} = v_{zonal} - 0.933\ v_{iz}$. This means that, for our experimental configuration, the vertical drifts in conjunction with the geometry of the signal path can contribute to the magnitude of the observed scintillation pattern velocities. Outside the pre-reversal enhancement (PRE) time that occurs around sunset, vertical drifts are expected to be much smaller than the values of zonal drifts. Following previous studies (e.g., Cesaroni et al., 2021), the effect of the vertical drifts are not corrected in our analyses.





## 4 Results and discussion

### 4.1 On the estimation of irregularity drifts velocities

We now present and discuss the measurements made by the UFCG ScintPi monitors and the zonal irregularity drifts derived
from these measurements. Figure 3 shows a representative example of the measurements and results. The measurements are
for the night between October 4 and 5, 2022.

Figure 3a shows L1 C/No values for the PRN136 signal measured by the two receivers. It shows that, as expected, similar
variations in scintillation occurrence and magnitude were observed in both signals.

Figure 3b shows 1-minute S4 index values computed for both signals. The S4 index is a widely index used in scintillation
studies to quantify amplitude scintillation (Groves et al., 1997; Kil et al., 2002; Nishioka et al., 2011) and can be defined as
the standard deviation of the signal intensity normalized by its average. The observed S4 values indicate (i) strong L-Band
scintillation occurring from 22:30 to 23:45 UT (20:30 to 21:45 LT) and (ii) moderate to weak scintillation between 00:30 and
02:30 UT (22:30 to 00:30 LT). Note that the estimates of S4 for both signals provide nearly identical values most of the time,
which indicates similar variability in the intensity patterns. This similarity in the signals is better quantified in the high cross-
correlation coefficients (i.e., $|r| \geq 0.75$) presented in Fig. 3c.

Figure 3c also shows that, despite the reduced resolution of the signal intensities, high correlation values can still be derived
from ScintPi measurements. Additionally, we point out that high cross-correlation coefficients persisted even during weak
scintillation activity (i.e., S4 in the 0.2 - 0.4 range), indicating that one can compute irregularity drifts even during periods of
weak L-Band scintillation.

Figure 3d presents the estimates of the characteristic velocity ($v_c$). The characteristic velocity has been averaged in 3-minute
bins (average of 3 values). The error bars represent the standard deviation of the values used in the averages. One can observe
that characteristic velocity shows higher values early in the evening. We attribute this to periods of more turbulent irregularity
motion associated with the development of EPBs. This turbulent or random behavior has been observed in previous studies
(e.g., Spatz et al.,1988; Bhattacharyya et al, 2001; Valladares et al., 2002; Kintner et al.,2004; Otsuka et al., 2006).

Finally, Fig. 3e shows our estimates (3-min averages) of the corrected true scintillation pattern velocities ($v_{scintx}$). Again,
error bars represent the standard deviation of the values used in the averages. As expected, the scintillation pattern velocities
reveal a behavior that is similar to that expected for the low latitude drifts. They show strong eastward drifts early in the
evening with weakening drifts towards midnight.




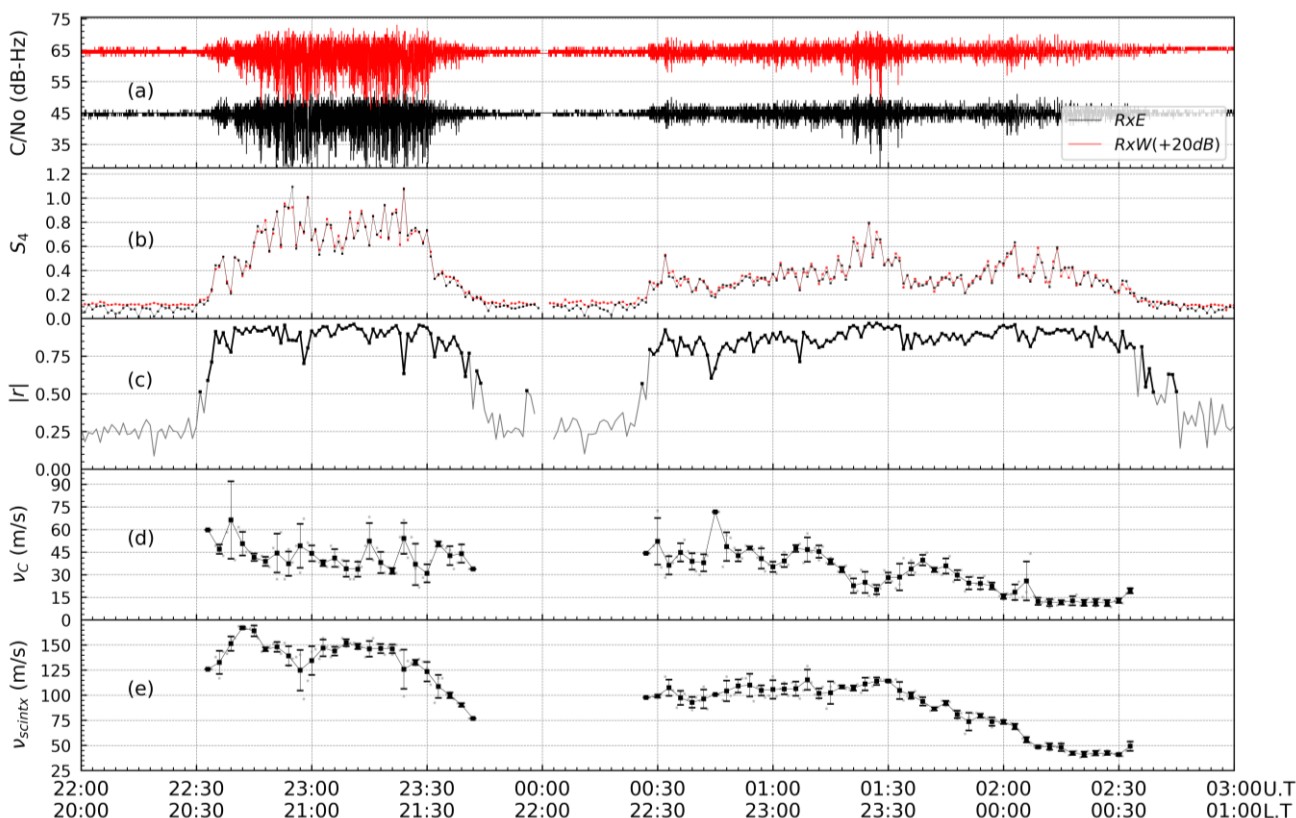

**Figure 3 - Example of measurements and analyses made with closely spaced ScintPi 3.0 monitors at UFCG for the same night of Fig.**

**2 (October 4-5, 2022). (a) Signal strenghts (C/No) in dB-Hz from L1 from PRN136. (b) Severity of scintillation, i.e., S4 index. (c) Maximum value from the normalized cross correlation function, highlighted in black any value greater than 0.5. (d) Estimated mean characteristic velocities, (e) Estimated mean true scintillation pattern velocities. Error bars in panels (d) and (e) represent standard deviation around the mean of the values used in 3-minute bins.**

## 4.2 On the seasonal variability of zonal irregularity drifts

The analysis described in Sect. 4.1 was applied to a data set containing 242 observation nights between September 1, 2022 and April 30, 2023. The period covers the equatorial spread F (ESF) season in Brazil (Abdu et al., 2003) when scintillation is observed. This period also has measurements from two equinoxes and December solstice. The results are summarized in Fig. 4.

Figure 4a shows the variation of the observed L-Band (1.575 GHz) scintillation as a function of local time and day of the campaign. The vertical white lines indicate missing data because of power outages at least one of the receiver sites. Figure 4a also shows the day-to-day variability of scintillation activity observed by the monitors. It also shows that scintillation



occurrence in most days. Figure 4a shows that L-Band scintillation was observed, mostly, between 20:00 LT and local midnight. During some days, however, scintillation occurred until 02:00 LT.


Figure 4b summarizes the magnetic zonal irregularity drift results. The drifts show a behavior that indeed is similar to what would be expected for background nighttime ionospheric plasma drifts at low latitudes (Fejer et al., 1991; Fejer et al., 2005). The drifts are predominantly eastward, with larger values (120 – 200 m/s) in the beginning of the evening and decreasing to lower values towards midnight. Eastward drifts around 20-80 m/s in the midnight and post-midnight sector. The results in

Figure 4c also show a trend, with lower drift values in September/October and April/May and higher values in the months between.

As mentioned earlier, zonal irregularity drifts are often assumed to be tracers of ionospheric plasma drifts (Ledvina et al., 2004). At low latitudes, theoretical analysis suggest that nighttime zonal drifts are expected to be controlled, in most part, by

zonal F-region winds (Rishbeth et al., 1972; Heelis et al., 1974). For instance, zonal plasma drifts are related to zonal neutral winds ($Uz$) through a flux-tube integrated Pedersen conductivity weighted integral which is representative of the neutral wind dynamo (Haerendel et al., 1992):

$$v_{zonal} \sim \frac{1}{\Sigma_P} \int \sigma_p U_z ds \tag{8}$$

Where the integral is performed along a magnetic field line, $\sigma_p$ is the Pedersen conductivity and $\Sigma_P$ is the Pedersen

conductance. At night, E-region conductivities are greatly reduced and zonal drifts are expected to have contributions coming, predominantly, from winds at F-region heights.

Figure 4c shows the magnetic zonal component of the neutral winds at F-region heights (450 km) over UFCG predicted by the Horizontal Wind Model 14 (HWM14). HWM14 is an empirical model of the neutral winds developed using an extensive set of measurements (Drob et al., 2015). It provides estimates of the geomagnetically quiet time behavior of the thermospheric

winds. Similar to the observed behavior of the irregularity drifts, Figure 4c shows stronger eastward winds in the beginning of the evening and weaker winds towards midnight and into the post-midnight sector. Additionally, HWM14 results show weaker zonal winds in September/October and April/May, contrasting with stronger zonal winds in the months between November and March. The similarity in the behavior between drifts and winds is impressive, given that the plasma drifts depend on combined contributions of winds along magnetic fields, weighted by the conductivity values. We must point out that

similarities in the behavior of EPB and neutral wind magnetic zonal velocities have been reported out before using, for instance, same-night observations made by airglow cameras and Fabry-Perot Interferometers (FPI) located near our site (UFCG) in Brazil (Chapagain et al., 2012).





**Figure 4 - Panel (a) shows the average S4 index measurements as a function of local time (LT = UT − 2) and date for 242 nights between September 1, 2022, and April 30, 2023. The average is for the two S4 values computed by the spaced monitors. White lines indicate lack of data caused by power outages at the observation sites. Panel (b) shows zonal irregularity drifts. Different seasons determined as +/- 45 days around equinox and solstice days are also indicated. Panel (c) shows climatological zonal wind velocities (*Uz*) from the Horizontal Wind Model 14 (HWM14) for a height of 450 km over UFCG, its values magnitude values are indicated by labels of the contour lines.**

We point out that the magnitudes of the winds and irregularity drifts differ, with irregularity drifts being, in general, larger than the zonal winds. The difference is attributed to the fact the drifts are the result of conductivity weighted winds along the magnetic field lines (Eq. 5) and the HWM14 winds shown here are for a single height and location (450 km above UFCG). Observations of neutral winds larger than ionospheric drifts have been reported in previous studies (e.g., Wharton et al., 1984; Coley et al., 1994; Navarro and Fejer, 2020). Additionally, HWM14 does not have a solar flux dependency and are

representative of the solar conditions under which the drift measurements were made.

Finally, we also point out cases of very low irregularity drift values early in the evening on certain observation nights. See, for instance, blue markers in some of the drift measurements between 19:00 and 20:30 LT in Fig. 4b. We hypothesize that these low values of zonal drifts can be caused by the contribution of large vertical drifts, associated with the PRE, to the estimates

of zonal drifts (Eq. 7). Estimates of nighttime vertical drifts derived from digisonde data for a low-latitude site in Brazil have revealed cases of upward drifts as large as 40 m/s during the same timeframe of 19:00 and 20:30 LT (e.g. Abdu et al., 2006).

## 5 Conclusions

We presented results of an experiment to evaluate the use of ScintPi 3.0 to estimate ionospheric irregularity drifts. ScintPi 3.0 is an alternative, low-cost GNSS-based scintillation monitor developed to enable research, education, and citizen science

initiatives (Gomez Socola and Rodrigues, 2022). Of relevance to this study is that ScintPi 3.0 is capable of measuring signals transmitted by geostationary satellites.

The experiment used the spaced-receiver scintillation technique for measurements of ionospheric irregularity drifts (Briggs et al., 1950; Ledvina et al, 2004). It also used measurements made by two ScintPi 3.0 deployed in Campina Grande, Brazil

(7.21°S, 35.91°W, dip latitude ~14°S). The monitors were spaced by 140 m in the magnetic east-west direction to measure the zonal component of the drifts associated with scintillation-causing equatorial spread F (ESF) irregularities.

We analyzed measurements made throughout an entire ESF season, between September 2022 and April 2023. We focused our analyses on irregularity drifts derived from geostationary satellite signals, which simplify the analyses and interpretation of

the results.

The results confirm that ScintPi 3.0, despite its low cost and low C/No resolution (1 dB-Hz), can measure irregularity drifts. The cross-correlation of the signals is mostly well above 0.75 even during late-night weak scintillation. Our drift calculation results also show that the local time variation in the estimated irregularity zonal drifts is in good agreement with previous

experimental studies and with theoretical expectations for the behavior of the low-latitude background zonal plasma drifts (e.g.,

Haerendel et al., 1992; Fejer et al., 2005; Shidler and Rodrigues, 2021). The drifts are predominantly eastward, with largest values early in the evening and weakening drifts towards local midnight.

Our results also reveal a seasonal trend in the irregularity zonal drifts. We found that this trend is well correlated with the behavior of the zonal component of the thermospheric neutral winds as predicted by the Horizontal Wind Model (HMW14). The similarity between the behavior of the neutral winds and observed irregularity drifts is a manifestation of the F-region dynamo, which controls the zonal F-region drifts at night (e.g., Haerendel et al., 1992).

**Data availability**

Ionospheric irregularity drift datasets are available at the link below. The dataset will be moved to a public repository (e.g., 345   Zenodo) after going through the review process.

(https://drive.google.com/drive/folders/1dHh8M1uBcI_8OzZfk69NrjwmFCkuDGua?usp=sharing)

**Author contribution**

JGS and FSR proposed the experiment and IP and RB host it at UFCG. JGS and IP oversaw the acquisition of data during the campaign. JGS performed the analyses with contributions from IW. JGS and FSR interpreted the results and prepared the 350   manuscript with contributions from all co-authors.

**Competing interests**

The authors declare that they have no conflict of interest.

**Acknowledgements**

Work at UTD was supported by the National Science Foundation (NSF) Award AGS-2122639 and by the NSF's GRFP Grant 355   No. 2136516. I. Paulino would like to thank Conselho Nacional de Desenvolvimento Científico e Tecnológico (CNPq) for supporting this research under contract 309981/2023-9.

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
