# Peer review of "ScintPi measurements of low-latitude ionospheric irregularity drifts using the spaced-receiver technique and SBAS signals"

_EGUsphere, 2024_

## Author Response (AR1)

**Responses to anonymous Referee #1**

Dear referee # 1, below you will find the answers to your questions and comments.

1. Line 105, could you use one sentence to summarize the relation between the spacing, irregularity drift, and sampling rate?

   As requested, we now provide in section 2.1 a sentence that summarizes the relation between spacing, irregularity drift, and sampling rate.

2. In Figure 3, do you use a threshold of |r| to select high correlation data points? If so, would this method be only applicable during strong scintillation conditions?

   Following the Referee#1's comment, we made changes in the text to make it clearer that the velocity estimation only uses scintillation patterns with |r| > 0.5. This threshold is now stated in the caption of Figure 3c.

   The method is not only applicable during strong scintillation conditions since selection is based on |r|. Even during moderate or weak scintillation, estimates of irregularity velocity can be made. This is how we obtain irregularity velocities near local midnight when L-Band scintillation is typically weak due to reduced background ionospheric densities. The reduction in scintillation severity late at night while |r| remains high can be seen in Figure 3.

   For completeness, section 4.1 states that "high cross-correlation coefficients persisted even during weak scintillation activity (i.e., S4 in the 0.2 - 0.4 range), indicating that one can compute irregularity drifts even during periods of weak L-Band scintillation."

3. In Figures 4b and c, while it seems that zonal drifts and winds show similarity in the local time and daily variation, it might be better to have a figure to show the correlation between them.

   We appreciate the comment by the referee#1. We initially considered to show individual curves and correlation values. We quickly realized that it would be misleading to show those since we would be comparing measurements against climatological models. The measurements have day-to-day variabilities and solar flux effects in them. HMW does not take into consideration solar flux. Additionally, the relationship between winds and ionospheric drifts is non-linear through the Pedersen conductivity weighted integral that takes into consideration the contribution of winds at different heights along magnetic field lines. Showing Figures 4(b) and 4(c) is the best compromise we found to those variabilities in local time and season.

4.  The IPP calculation considers the altitude of the ionospheric F-region peak density at 450 km. How do different choices of this value impact the estimation of drift velocities? It might be better to have some discussion or background literature on this.

    This highlights the benefits of using signals from geostationary satellites. Values of the F-region peak density in a realistic range (say, 300 to 600 km) have minimal impact on the estimation of drift velocities. Of relevance for our calculation are the magnetic field vector components, $B_x$, $B_y$, and $B_z$ at the IPP location, which do not change much for typical F-region heights. In fact, this characteristic was leveraged to estimate the scattering height of irregularities by comparing velocities derived from geostationary and moving satellites, as demonstrated by Cerruti et al., 2006.

    Following the referee#1's suggestion, we added clarification about the impact of F-region height in the drift calculation in Section 3.1.

Minor comments:

1.  Line 267-268, please check "It also shows that scintillation occurrence in most days".

    Corrected. Thank you.

2.  Line 274, please check "Eastward drifts around 20-80 m/s in the midnight and post-midnight sector".

    Corrected. Thank you.

3.  Line 180, is there any reason to use west receiver instead of east receiver?

    The west receiver seems to be less susceptible to multipath.

4.  Line 216, could you explain more for the two values in θ?

    The primary reason is that θ is the azimuth angle of the geostationary satellite with respect to magnetic north (definition earlier on in Section 3.1). The magnetic declination for our setup is -21.39 deg. and the geographic azimuth angle of the geostationary satellite is 84 deg,. Therefore, θ = 84 + 21.39 = 105.39 deg. To avoid confusion, we now only provide the final θ value, 105.39 deg.

**Responses to anonymous Referee #2**

Dear referee#2, below (in blue) you will find the answers to your questions and comments.

1. In equation (6), a value of 0.899 was obtained. I suggest including the values used to reach this result, perhaps in a table.

   We agree that showing these values clarity our calculations. Following your suggestion we now have specify these values in equation 6 and remind the reader that the variables are described Section 2.2.

2. Was a threshold value applied to define a high correlation |r|? Figure 3c suggests that this threshold might be 0.5. If so, this should be explicitly stated in the manuscript

   In the caption of Figure 3, we have made more explicit the threshold of |r| > 0.5. Additionally, in Section 4.1, we also add an explanation that these correlation values can be obtained even during weak scintillation conditions.

3. The altitude of the ionospheric F-region peak density is mentioned as 450 km several times in the manuscript. Why were 450 km chosen? If there is a specific reason, please provide justification in the text

   While 350 km is often used, there are studies showing that the optimal F-region height used for IPP calculation can vary between 400 km and 500 km for low and mid latitude regions (Nava et al., 2007). Also, there are studies of hmF2 that show pre-midnight values around 450 km (e.g., Abdu et al., 2006) for low latitude stations. The IPP was computed using a mean ionospheric altitude of 450 km that is thought to be adequate for the location and solar flux conditions of the observations. More importantly, different choices of ionospheric altitude (say, 300 to 600 km) do not affect the drift estimates and results. Following the referee#2's comments, these are now better explained in Section 3.1.

   References:

   Abdu, M. A., Batista, I. S., Reinisch, B. W., Sobral, J. H. A., & Carrasco, A. J. (2006). Equatorial F region evening vertical drift, and peak height, during southern winter months: A comparison of observational data with the IRI descriptions. Advances in Space Research, 37(5), 1007-1017.

   Nava B, Radicella SM, Leitinger R, Coïsson P (2007) Use of total electron content data to analyze ionosphere electron density gradients. Adv Space Res 39(8):1292–1297. doi:10.1016/j.asr.2007.01.041